



# Lagoon hydrodynamics of pearl farming islands: the case of Gambier (French Polynesia)

Oriane Bruyère[1], Romain Le Gendre[2], Vetea Liao[3], Serge Andréfouët[1,4]

[1]IRD, UMR 9220 ENTROPIE (IRD, Univ. La Réunion, IFREMER, Univ. Nouvelle-Calédonie, CNRS), BPA5, 98948 Nouméa, New Caledonia

[2]1Ifremer, UMR 9220 ENTROPIE (IRD, Univ. Réunion, IFREMER, Univ. Nouvelle-Calédonie, CNRS), BP 32078, 98897 Nouméa CEDEX, New Caledonia

[3]Direction des Ressources Marines, BP 20, 98713 Papeete, French Polynesia

[4]IRD, UMR-9220 ENTROPIE (Institut de Recherche pour le Développement, Université de la Réunion, IFREMER, CNRS, Université de la Nouvelle-Calédonie), BP 49, 98725 Vairao, Tahiti, French Polynesia

*Correspondence to*: Serge Andréfouët (serge.andrefouet@ird.fr)

## Abstract.

Between 2019 and 2020, the Gambier lagoon was instrumented over a period of 9 months with a large array of autonomous oceanographic instruments measuring temperature, pressure and current. Two deployments were conducted, respectively from June 2019 to October 2019 (Leg1) and from late October 2019 to late February 2020 (Leg2). A total of sixteen instrumented locations were spread across the lagoon and on the forereef. Physical parameters were measured to characterize the wave climate, tide and surges, lagoonal circulation, and spatial and vertical temperature variabilities. Those observations were part of the ANR-funded MANA project (2017-2022) and its derivatives that aimed to improve knowledge of processes influencing the spat collection of *Pinctada margaritifera* oysters that are used for black pearl farming and production. This data set was a prerequisite for the development of a high resolution biophysical model on Gambier lagoon which aims at tackling the connectivity oyster larvae. The sampling strategy focused on the northern region of Gambier lagoon and especially on the sub-lagoon of Rikitea which is a prime spat collection site. The data set was post processed, quality controlled and is archived in a dedicated repository with a permanent DOI into the SEANOE marine data platform.

## 1 Introduction

Black pearl farming is the second major economic income for French Polynesia representing about 40M€ of international sales. The sector employed 1300 workers on 28 atolls and islands as in 2021 and contributes to stabilize populations in remote islands especially in the Tuamotu and Gambier





archipelagoes. The Gambier Islands, are a series of small volcanic islands within a single large lagoon
representing 25% of French Polynesia black pearl production in 2020 (André et al., 2022, Bruyère et al.,
2023a). This site has been however rather overlooked in term of scientific investigations aiming to
support pearl farming management. Despite its good pearl production performances, Gambier also
suffers from problems, such as lagoonal space limitation (André et al., 2022) and more critically a

recent decrease in spat collection rates (Bruyère et al., 2023a). Better understanding of spat collection
require the characterization of the oyster stocks (Bionaz et al., 2022), oyster life cycle (Le Moullac et
al., 2012), and a good knowledge of hydrodynamic functioning of lagoons in relation to forcing factors
such as tides, wind or waves. Several pearl farming Tuamotu atoll lagoons have been instrumented in
the past to study their hydrodynamics (Dumas et al., 2012; Andréfouët et al. 2023a; Bruyère et al.,

2023b) and Gambier was added to the list starting in late 2019 as the first high volcanic islands
investigated with an array of physical oceanography instruments (Bruyère et al., 2023a). The sampling
strategy is different than the general strategy applied to atolls as the focus of the investigation on
Gambier was on a spat-producing sub-lagoon (Bruyère et al., 2023a) and because the *hoa* (vernacular
name for the shallow passages transversal to an atoll rim) were not instrumented here unlike for atolls.

As such, this paper presents the hydrodynamic data recorded in Gambier lagoons during a 9-month
deployment period (June 2019-February 2020). Similar data collected on atolls are presented elsewhere
(Bruyère et al., 2023b).

## 2 Study Site

Gambier Islands (23°07′ S – 134° 58′ W) are a group of seven high islands sharing the same large deep
lagoon, situated 1645 km South-Eastward from Tahiti Island. It is part of the administrative Gambier
archipelago (Figure 1) and counts 1 592 inhabitants in 2017. The Gambier lagoon is very open to the
ocean, being surrounded by a barrier reef that is submerged on the southern and western part, and by an
intertidal and emerged barrier reef in the north and east sides (Figure 2). The north side is cut in some

sections by several *hoa*, allowing connections between the lagoon and the deep ocean.

Accurate bathymetric data at high spatial resolution are needed to model the Gambier lagoon
hydrodynamic with realism. The French Hydrographic and Oceanographic Service (SHOM) previously
collected point bathymetric soundings at high density but not everywhere in the lagoon. To fill the gaps,

the *Direction des Ressources Marines* (DRM) of French Polynesia has funded in 2020 a multi-beam
bathymetric survey wherever a small vessel could navigate. To fill the last gaps remaining in the
northern shallow lagoon and south-western shallow reef flats, all available *in situ* soundings trained a
satellite-derived bathymetry computed from a 10-meter spatial resolution Sentinel-2 MSI MultiSpectral
Instrument imagery following the method described by Amrari et al. (2021). For the hydrodynamic

model, a final bathymetric grid at 100m resolution was produced by merging the *in situ* and satellite-
derived bathymetry data and resampling it at 100m resolution (Figure 2). This model resolution was
deemed suitable for our purpose following our experience in atoll model development, but it is also
confirmed by recent sensitivity analyses study in other coastal environments (Ward et al., 2023).





Previous bathymetric work using Landsat MSS noisy data at 80m spatial resolution was described by
Pirrazoli et al. (1984) but data are not available and the results could not be evaluated or used.

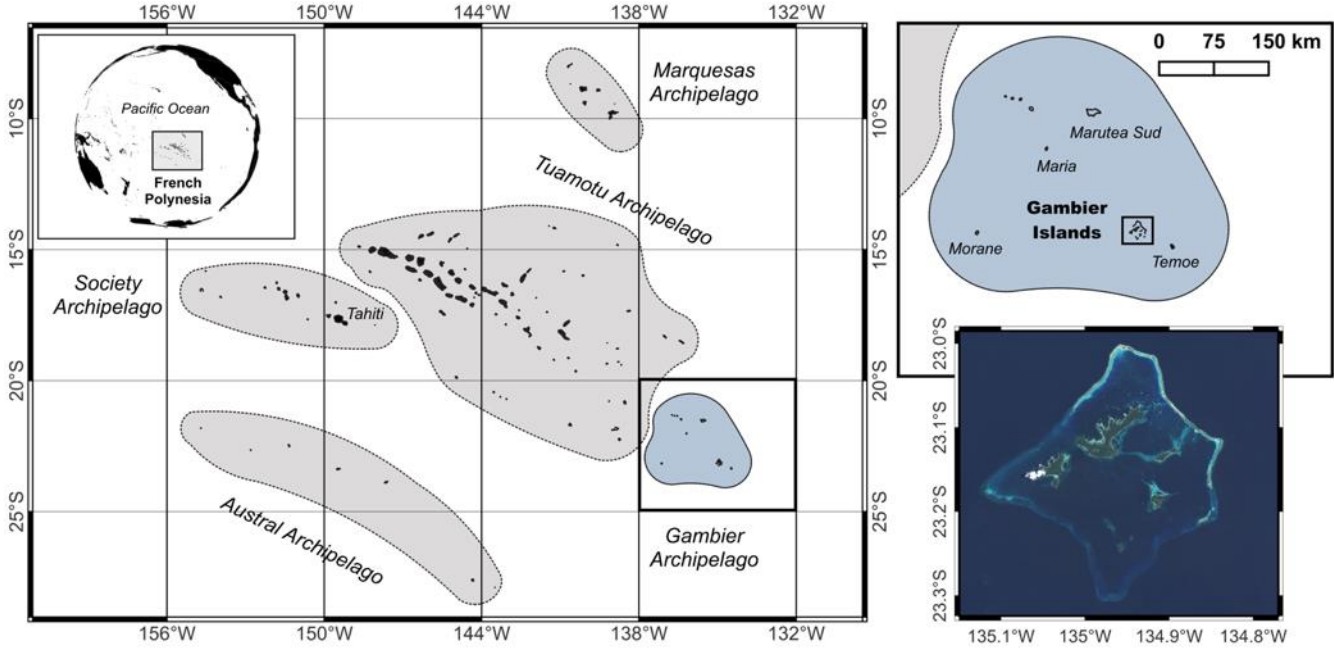

**Figure 1: Map of French Polynesia archipelagos and zoom of Gambier Islands location. Background satellite image from Sentinel-2 mission, European Space Agency (ESA).**

The Gambier lagoon has a surface area of approx. 500 km². Mean and maximum are 24.5m and 71m
deep respectively. The lagoon is geomorphologically complex with several deep basins, deep reticulated
structures, patch reefs at various depths, and shallow sills and reef flats. Of interest is the Rikitea sub-lagoon that faces on its west side the Rikitea Village of Mangareva (main inhabited island). It is
bordered on the south by Aukena Island and a shallow reef flat, and on the north side by several large
pinnacles separated by deep channels (Figure 2, insert). Finally, its eastern part is bounded by the semi-continuous emerged barrier reef. A deep basin reaching 70m depth is present in the central region of
Rikitea lagoon. This sub-lagoon is a priority study site because it is the main spat collecting site for
local pearl farmers.





**Figure 2: Bathymetry map of Gambier lagoon at 100m resolution. The inset shows the Rikitea sub-lagoon, a key spat collection location for local pearl farmers.**

## 2.1 Meteorological conditions

Meteorological conditions are recorded by Météo France weather station which is set on the southeast of Mangareva Island at 91 altitude meters. It measured wind speeds and directions at hourly time steps.





However, the station can be influenced by local orographic effects (Laurent and Maamaatuaiahutapu, 2019). For this reason, we decided to use reanalysis data from ERA5 model (Hersbach et al.; 2020)(Figure 3), data were extracted in a single location inside the lagoon (23°09'11.6"S - 134°57'31.4"W) (Figure 4).


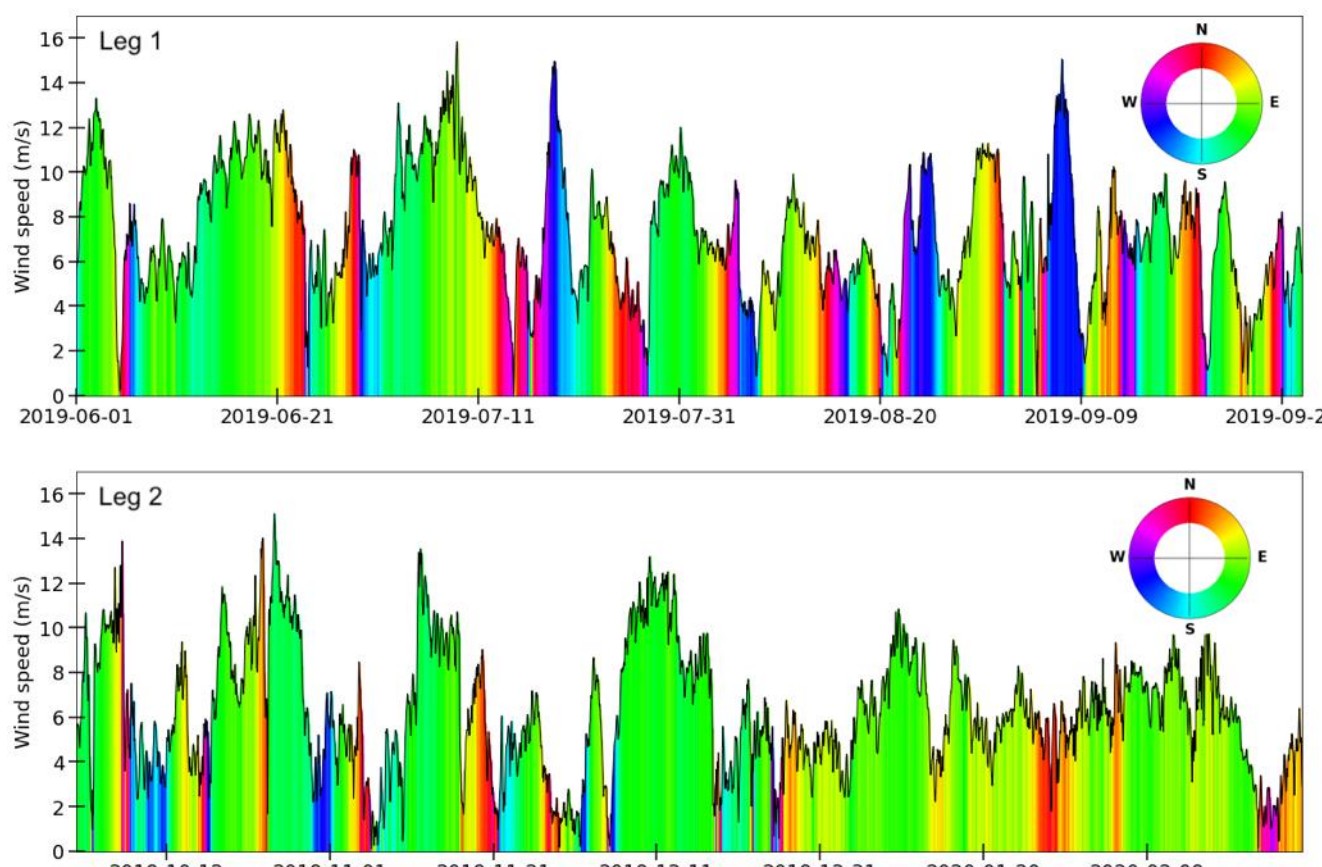

**Figure 3: Wind speeds (black line) and directions (polar legend) from ERA5 reanalysis were extracted inside Gambier lagoon during Leg 1 and Leg 2 deployments. Wind directions follow meteorological convention.**


Wind conditions observed during Leg 1 are dominated by strong and short periods of SW winds probably related to wave events. Conversely, Leg 2 is dominated by trade wind (E to SE winds) with several SE winds events above 10 m.s$^{-1}$.

## 3 Sampling Strategy

The sampling strategy mostly focused on the northern part of the Gambier lagoon, and especially the Rikitea lagoon (Figure 4). The sampling aimed to characterize the inward-outward flows from the three





boundary sectors, in the north, south-east and south orientations (Bruyère et al., 2023a). Beyond the Rikitea lagoon, the sampling also targeted the different sections of the barrier reef with a different exposition in order to characterize the incoming waves. Eventually, sixteen locations (or stations) were
equipped with at least one oceanographic instrument measuring temperature (SBE56), temperature/pressure (RBRduet T.D), currents direction and speed (ADCP, Aquadopp and Marotte HS). Instruments were moored between June 2019 and February 2020.

Observations were separated into two distinct legs (Leg1 from June 2019 to October 2019 and Leg2
from October 2019 to February 2020). At the beginning of Leg 1, the sampling strategy included:

- To measure the vertical temperature variability and potential stratification, four lagoon stations (L01, L02, L03, L04) are coupled with three instruments (two SBE56 and one RBRduet T.D): the SBE56 were moored at approximately 20m and 2m depth and the RBRduet T.D at mid-depth (7-
8m). The spatial replication in four stations allowed to measure the lagoon heterogeneity.
- To measure the incident wave parameters, five RBRduet T.D were positioned (Significant wave height, Mean wave period 01 and Peak frequency), water elevation, surges and temperature in 5 locations (O01 to O05) on outer forereef sections each with a different orientation/exposure. Instruments were moored in about 10 meters of water.
- To monitor current speeds and directions, three low-cost Marotte HS loggers were deployed in P01 to P03 on shallow reef flats. P02 and P03 monitored water entries into and from the Rikitea lagoon through Mangareva and Aukena reef flat passage (P02), and in the north of Aukena Island (P03).

All instruments were retrieved at the end of Leg 1 in late October 2019, allowing to download data,
changes batteries, clean probes and check mooring component. At the beginning of Leg2, this initial set-up was enlarged with six current profilers (4 ADCPs and 2 Aquadopp) to measure current speeds and directions in strategic locations, namely the three open sides of the Rikitea sub-lagoon (ADCP_1 in the North, Aqua1 in the south and Aqua 2 in the east), the western deep pass (ADCP_3), the south channel (ADCP_4) and a location of the western lagoon close to farming sites (ADCP_2). The two
Aquadopp were coupled with Marotte HS loggers previously moored in P02 and P03 shallow stations to compare the agreement between records from different sensor types and to provide more robust observations with profiles of the water column.

A detailed list of moored instruments is presented in Table 1 detailing instruments model, geographic
positions, depth of deployment, sampling frequencies, dates of measurement and measured physical parameters.

Earth System
Science
Data

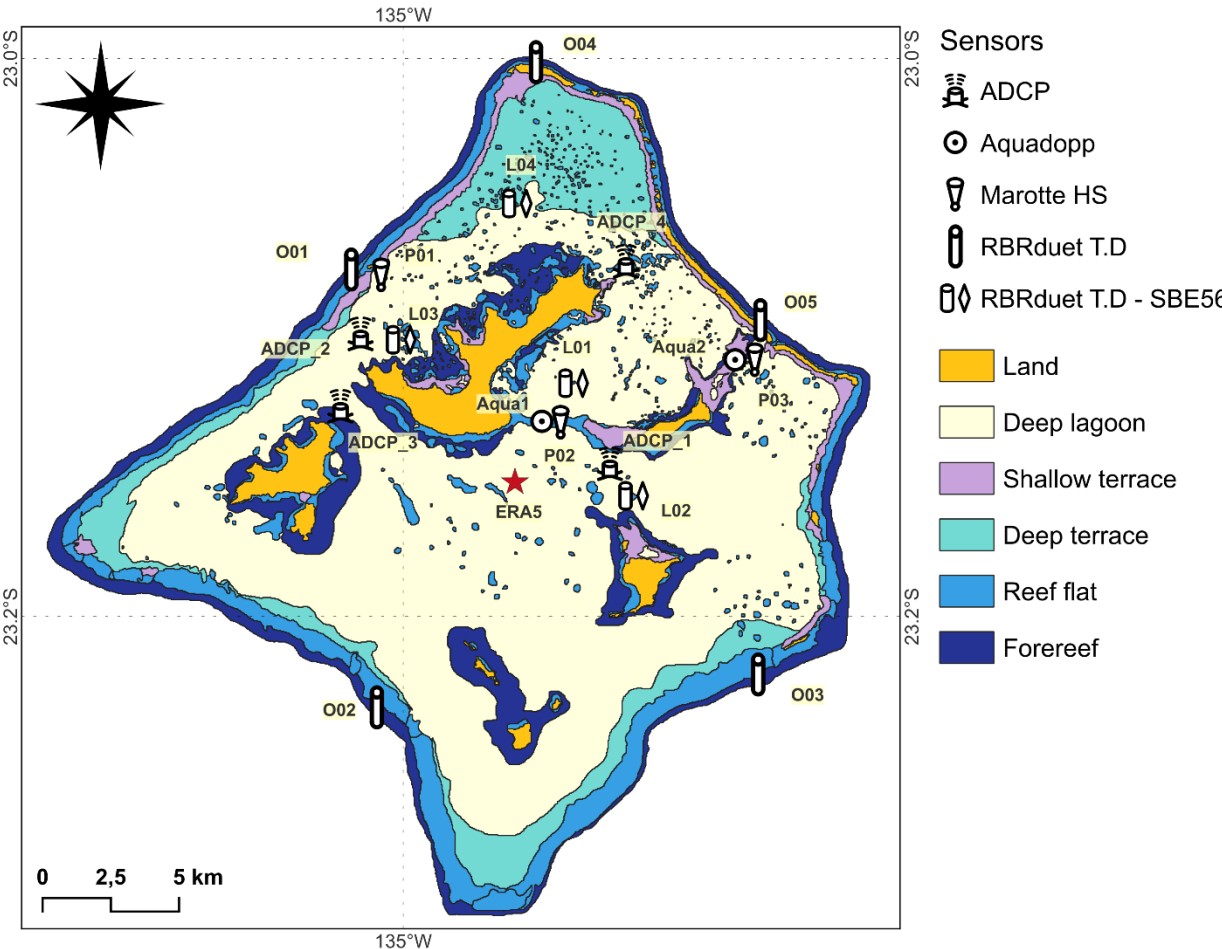

**Figure 4: Sampling strategy deployed during Gambier surveys. ADCP: Acoustic Doppler Current Profiler. Background map from**
**the Millennium Coral Reef Mapping Project (Andréfouët and Bionaz, 2021). The red star represents the point for which ERA5**
**reanalysis model data are extracted. .**

**Table 1: Detailed list of instruments moored into Gambier lagoon.**

| Station | Instrument | Raw parameters | Longitude (W) | Latitude (S) | Date Start | Date End | Freq | Depth (m) | Processed parameters | Legs |
|---|---|---|---|---|---|---|---|---|---|---|
| GAMBIER ISLANDS | | | | | | | | | | |
| O05 | RBRduet T.D | Temperature – pressure | 134.87201 | 23.09379 | 15/06/2019 | 24/02/2020 | 1 Hz | 9.4 | Temperature – wave height & period – water level | 1,2 |
| O04 | RBRduet T.D | Temperature – pressure | 134.95241 | 23.0013 | 15/06/2019 | 24/02/2020 | 1 Hz | 11 | Temperature – wave height & period – water level | 1,2 |
| O03 | RBRduet T.D | Temperature – pressure | 134.87282 | 23.2208 | 15/06/2019 | 23/02/2020 | 1 Hz | 12.6 | Temperature – wave height & period – water level | 1,2 |



| O02 | RBRduet T.D | Temperature – pressure | 135.00941 | 23.23259 | 15/06/2019 | 23/02/2020 | 1 Hz | 13 | Temperature – wave height & period – water level | 1,2 |
|---|---|---|---|---|---|---|---|---|---|---|
| O01 | RBRduet T.D | Temperature – pressure | 135.01859 | 23.07563 | 15/06/2019 | 24/02/2020 | 1 Hz | 12.2 | Temperature – wave height & period – water level | 1,2 |
| L01 | RBRduet T.D | Temperature – pressure | 134.93874 | 23.11633 | 15/06/2019 | 26/02/2020 | 1 Hz | 7.4 | Temperature – wave height & period – water level | 1,2 |
| L01 | SBE56 | Temperature | 134.93892 | 23.1164 | 15/06/2019 | 26/02/2020 | 10 min | 3 | Temperature | 1,2 |
| L01 | SBE56 | Temperature | 134.93819 | 23.11623 | 15/06/2019 | 26/02/2020 | 10 min | 21 | Temperature | 1,2 |
| L02 | RBRduet T.D | Temperature – pressure | 134.91771 | 23.15682 | 15/06/2019 | 23/02/2020 | 1 Hz | 7 | Temperature – wave height & period – water level | 1,2 |
| L02 | SBE56 | Temperature | 134.91864 | 23.15641 | 15/06/2019 | 23/02/2020 | 10 min | 4 | Temperature | 1,2 |
| L02 | SBE56 | Temperature | 134.91739 | 23.15676 | 15/06/2019 | 23/02/2020 | 10 min | 21 | Temperature | 1,2 |
| L03 | RBRduet T.D | Temperature – pressure | 135.0009 | 23.10086 | 15/06/2019 | 23/02/2020 | 1 Hz | 8.3 | Temperature – wave height & period – water level | 1,2 |
| L03 | SBE56 | Temperature | 135.00069 | 23.10083 | 15/06/2019 | 23/02/2020 | 10 min | 3 | Temperature | 1,2 |
| L03 | SBE56 | Temperature | 135.00121 | 23.10088 | 15/06/2019 | 23/02/2020 | 10 min | 23 | Temperature | 1,2 |
| L04 | RBRduet T.D | Temperature – pressure | 134.95927 | 23.05208 | 15/06/2019 | 24/02/2020 | 1 Hz | 7.8 | Temperature – wave height & period – water level | 1,2 |
| L04 | SBE56 | Temperature | 134.95937 | 23.05204 | 15/06/2019 | 24/02/2020 | 10 min | 3 | Temperature | 1,2 |
| L04 | SBE56 | Temperature | 134.95912 | 23.052 | 15/06/2019 | 24/02/2020 | 10 min | 22 | Temperature | 1,2 |
| P01 | Marotte HS | Currents - Temperature | 135.01491 | 23.07758 | 08/06/2019 | 12/10/2019 | 1 min | 5 | Temperature – current speed & direction | 1,2 |
| P02 | Marotte HS | Currents - Temperature | 134.95062 | 23.13042 | 10/06/2019 | 26/02/2020 | 1 min | 2 | Temperature – current speed & direction | 1,2 |
| P03 | Marotte HS | Current - Temperature | 134.88107 | 23.10799 | 10/06/2019 | 24/02/2020 | 1 min | 3 | Temperature – current speed & direction | 1,2 |
| ADCP_1 | ADCP Sentinel V50 | Current – pressure - temperature | 134.92593 | 23.14531 | 30/10/2019 | 19/12/2019 | 20 min | 35 | Temperature – current speed & direction – water level | 1 |
| ADCP_2 | ADCP Sentinel V20 | Current – pressure - temperature | 135.01524 | 23.09875 | 30/10/2019 | 24/02/2020 | 10 min | 19 | Temperature – current speed & direction – water level | 1 |
| ADCP_3 | ADCP Sentinel V20 | Current – pressure - temperature | 135.02245 | 23.12469 | 30/10/2019 | 23/02/2020 | 10 min | 16.6 | Temperature – current speed & direction – water level | 1 |
| ADCP_4 | ADCP Sentinel V50 | Current – pressure - temperature | 134.92017 | 23.07273 | 30/10/2019 | 24/02/2020 | 20 min | 31.4 | Temperature – current speed & direction – water level | 1 |
| Aqua1 (~P02) | Aquadopp Nortek | Current - pressure | 134.9503 | 23.13006 | 31/10/2019 | 26/02/2020 | 10 min | 8.6 | Current speed & direction – water level | 1 |



| Aqua2 (~P03) | Aquadopp Nortek | Current - pressure | 134.88107 | 23.108 | 31/10/2019 | 24/02/2020 | 10 min | 3.6 | Current speed & direction – water level | 1 |
|---|---|---|---|---|---|---|---|---|---|---|

## 4 Instruments, Data processing and Quality control

Hereafter, we present the five autonomous coastal oceanographic instruments measuring currents, temperature and pressure used for Gambier campaigns. Instruments were moored by SCUBA on the sea floor and secured on dedicated structure ensuring the data logger stability. Compact loggers were placed inside PVC cylinders and current profilers were protected with electrical tape to ease the removal of biological fouling. The settings were a compromise between measurements range and accuracy in the
deployment environment.

Raw data were downloaded using manufacturer's software, and processed with Python routines in order to generate NetCDF files. Global Attributes in NetCDF files provide details about the station (depth, geospatial coordinate), instrument settings (sampling frequency, serial number), contacts and project
references as well as any necessary additional comments useful for data users. However, specific processing steps required for some data sets are presented hereafter.

### 4.1 RBRduet T.D

RBRduet T.D sensor (RBR Ltd) is a compact logger ideal for long term deployments and providing
measurements of temperature and pressure at high frequency. This instrument was moored on external reef slopes or inside the lagoon. Five loggers were anchored between 9 and 13m depth in five distinct sides of Gambier Islands (north east, north, north-west, south-west and south-east) to measure the incident waves reaching the reef crest. Post processing provides Significant Wave Height, Mean Wave Period, and Peak Frequency. Inside the lagoon, four RBRduet T.D were moored around 7-8m to
measure the water level and to deduce surge signal. For each station, data logger was set-up to measure at 1Hz interval.
Pressure data were corrected from a constant atmospheric pressure value set to 101 325 bar, then burst data were filtered using the Fourier analysis to get a pressure spectra with a frequency period evaluated between 3 and 25s. The sea surface elevation spectra were processed using the linear theory and the
corrected pressure data to retrieve the waves parameters. Two processed files are generated, one file at hourly time step including wave parameters, water level and temperature and another at 1-minute frequency with water level and temperature.

### 4.2 SBE56

High accuracy (± 0.002 °C) temperature data were recorded with SBE56 sensors designed by
SEABIRD Electronics Inc. Instruments. These sensors were placed in each lagoon station at approx. 2m and 20m depth in order to detect vertical temperature stratification. The start time and configuration of



all instruments were identical. The measurement interval was set to record every 1-minute. Raw data did not need any processing stage and were directly converted into NetCDF file.

## 4.3 Current Profilers

ADCPs used during the Leg2 are of two types: ADCP Sentinel V from Teledyne RD Instruments Inc. (TRD-I) and Nortek Aquadopp current profiler with pressure sensor. Both instruments were bottom mounted and faced upward.

- Regarding ADCPs, the two working frequency models Sentinel V50 (500kHz) and Sentinel V20
(1000kHz) were used to measure velocity along the water column, and temperature and pressure at the sensor in its transducer head. ADCPs were set to measure in bust time: the V20 models measured burst each 10 minutes with 40 pings per ensemble. Conversely, the V50 models recorded every 20 minutes with 180 pings per burst. For both models, cell size was fixed to 1m resolution. The ADCPs V20 were moored at 17 and 19m depth and the V50s were fixed deeper, at 31 and 35m
deep.
- Aquadopp current profilers (2MHz version) measured three-component (east, north, up) of current velocity data in shallow areas. The instrument settings were set to measure at 600s burst interval (10min), each burst made of 3 pings. It resolved the entire water column with a cell size of 50cm. The ENU coordinate system was systematically used. Instruments were anchored at 3.6 and 8.6m
depth.

For the current profilers, the near sea-surface cells contaminated by the acoustic sidelobe reflections were removed. Valid bins were then used to retrieve currents magnitude and direction calculated using the zonal ($u$) and meridional ($v$) components. Pressure data were converted into depth by subtracting the
depth-averaged value across the entire time series to pressure. No barometric correction was applied to depth. Processed profiles data were eventually converted into NetCDF files.

## 4.4 Marotte HS

The Marotte HS drag-tilt current meters manufactured by Marine Geophysics Laboratory of James Cook University allow to measure temperature and $uv$ components at the level of the instrument within
a range of adequate conditions. Each Marotte HS used in Gambier were moored on a bottom structure and set to sample at 1 minute interval. With Python routines, vectors data ($u,v$) are converted in Clockwise from North convention to be consistent with oceanographic convention, then speed and direction are deduced. Note that the Marotte HS moored in P01 was lost during Leg 1.

## 4.5 Quality control

The quality control procedure consists of visually checking each time series converted in NetCDF files with Python and Ferret and remove all remaining out of water data, or flag anomalies (such as out of range data or spikes). With this step, the final correct date coverage is determined and reported in the Global attributes of each processed files.



# 5 Hydrodynamic overviews

This section briefly presents the hydrodynamic features of Gambier lagoons. Measurements have provided water level variations in the ocean and in the lagoon, incident waves on forereefs, spatial and vertical temperature variations, and finally the depth-averaged current at a number of strategic locations. Data shown here are complementary to those shown in Bruyère et al. (2023a) which were more specific to the study of the Rikitea sub-lagoon.


## 5.1 Tidal analysis

Amplitude of tidal constituents are presented in Table 2. Tidal analysis on Leg1 was made with the 't_tide' python package. The sum of the six main harmonic constituent reaches 48cm in oceanic station (O01). For O01 station, 57% of water elevation is related to the semi-diurnal M2 harmonic amplitude

which represents 27.7cm. Water level oscillations in ocean and lagoon are similar. (Figure 5). Compared to previous works on Tuamotu atolls (Dumas et al., 2012; Andréfouët et al., in review) where lagoon tide signal is highly attenuated, only few centimeters difference in amplitude occur in Gambier lagoon. It was less than 1cm between O01 and L01 stations but more than 2.6cm between O05 and L03) (Table 2). Those results confirm the degree of openness of Gambier lagoon to the ocean, and explain the

tidal driven circulation inside lagoon.

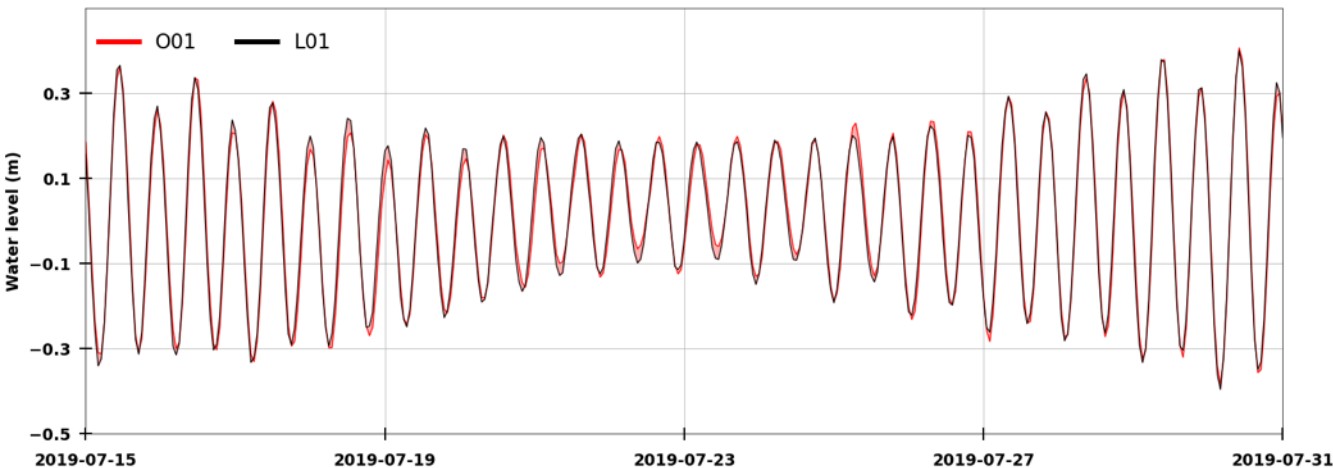

**Figure 5: Tide signals recorded with RBRduet T.D on forereef (station O01) and inside lagoon (station L01) during Leg1.**




**Table 2: Comparison of principal harmonic amplitude (cm) measured during Leg1 between oceanic station (O01, O02, O03, O04, O05) and lagoonal stations (L01, L03, L02, L01).**

| | Oceanic Stations | | | | | Lagoonal Stations | | | |
|---|---|---|---|---|---|---|---|---|---|
| Harmonic | O01 | O02 | O03 | O04 | O05 | L01 | L02 | L03 | L04 |
| M2 | 27.7 | 24.98 | 27.84 | 28.66 | 29.98 | 26.77 | 26.79 | 27.34 | 27.35 |
| S2 | 8.0 | 8.47 | 8.61 | 8.29 | 8.51 | 8.52 | 9.09 | 8.29 | 8.44 |
| N2 | 7.84 | 6.64 | 7.41 | 7.83 | 7.93 | 7.23 | 7.12 | 7.63 | 7.54 |
| K1 | 2.96 | 2.67 | 2.53 | 2.78 | 2.64 | 2.59 | 2.23 | 2.87 | 2.75 |
| O1 | 1.28 | 1.83 | 1.59 | 1.28 | 1.39 | 1.70 | 1.79 | 1.45 | 1.55 |
| Q1 | 0.35 | 0.50 | 0.37 | 0.27 | 0.26 | 0.45 | 0.49 | 0.38 | 0.42 |
| Total | 48.13 | 45.09 | 48.35 | 49.11 | 50.71 | 47.26 | 47.51 | 47.96 | 48.05 |

## 5.2 Wave parameters

Waves around Gambier were recorded on the five forereef equipped with the RBRduet T.D sensors. Leg1 deployment occurred during austral winter season (June to October) where distant swells generated by southern weather events are dominant. The measurements confirmed the importance of these southern swells (Figure 6A). The station O03 located on the south-eastern side of Gambier Islands recorded several events with wave heights >3m. Conversely, the significant wave heights measured on
the northern station O04 averaged at only ~1m excepted during few events at ~2m (i.e., July 2019). Only the two southern stations (O02 and O03) captured waves above 3m height (Figure 6B). The north-west side is the calmest area due to its protection from E-SE trade winds, with wave height under 1m in 73% of the time. For O03 station this class of height (0-1m) represented only 6.6% of the records.

Stations O04 (North) and O05 (North-east) recorded in ~80% of the time peak wave periods (Tp) under 11s. These stations are dominated by wind waves generated by E-NE trade winds. In contrary, O01, O02, O03 measured higher Tp which means that wave regimes are rather dominated by distant swell (Figure 6C).

Wave measurements during the Leg2 deployment in Austral summer are quite contrasted with Leg 1 (Figure 7A), in particular the wave heights measured on the south sides of the island (O02 or O03) reached only once 3m height (Figure 7B). Wave heights between 1-2m were dominant (> 50% of waves) on the southern stations (O02 and O03) and on the eastern station (O05). While, peak period broadly matched the winter season (Figure 7C).



Figure 6: (A) Time series of Significant Wave Height (Hsig) recorded at one hour time step for O03 (South-East side) and O04 (North side) stations during Leg1. (B) Percentage of occurrence (%) of wave height class between 0 and 6m height, measured on the five oceanic stations from 16th June to 1st October 2019. (C) Percentage of occurrence of Peak Period class between 6 to 22s.


Earth System Science Data Discussions · Open Access



**Figure 7: (A)** Time series of Significant Wave Height (Hsig) recorded at one hour time step for O03 (South-East side) and O04 (North side) stations during Leg2. **(B)** Number of occurrences (%) depending on wave height class (between 0 and 6m height) measured on the five oceanic stations over the period from 2$^{nd}$ September 2019 to 23$^{rd}$ February 2020. **(C)** Number of waves (%) according to Peak Period class (6 to 22s).



Based on Figure 8, wave events recorded during Leg1 on the southeast forereef station (O03) are also observed inside the lagoon (L02 station), but with a reduction in wave amplitude likely due to wave energy dissipation on the subtidal reef flats in the south (bottom friction), and the wave refractions on islands and very shallow or emerged reefs. The lagoon surge amplitude (L02) reacts in the same manner than surge measured in oceanic station (O03). This weak difference shows that the water levels are homogenous between lagoon and ocean.

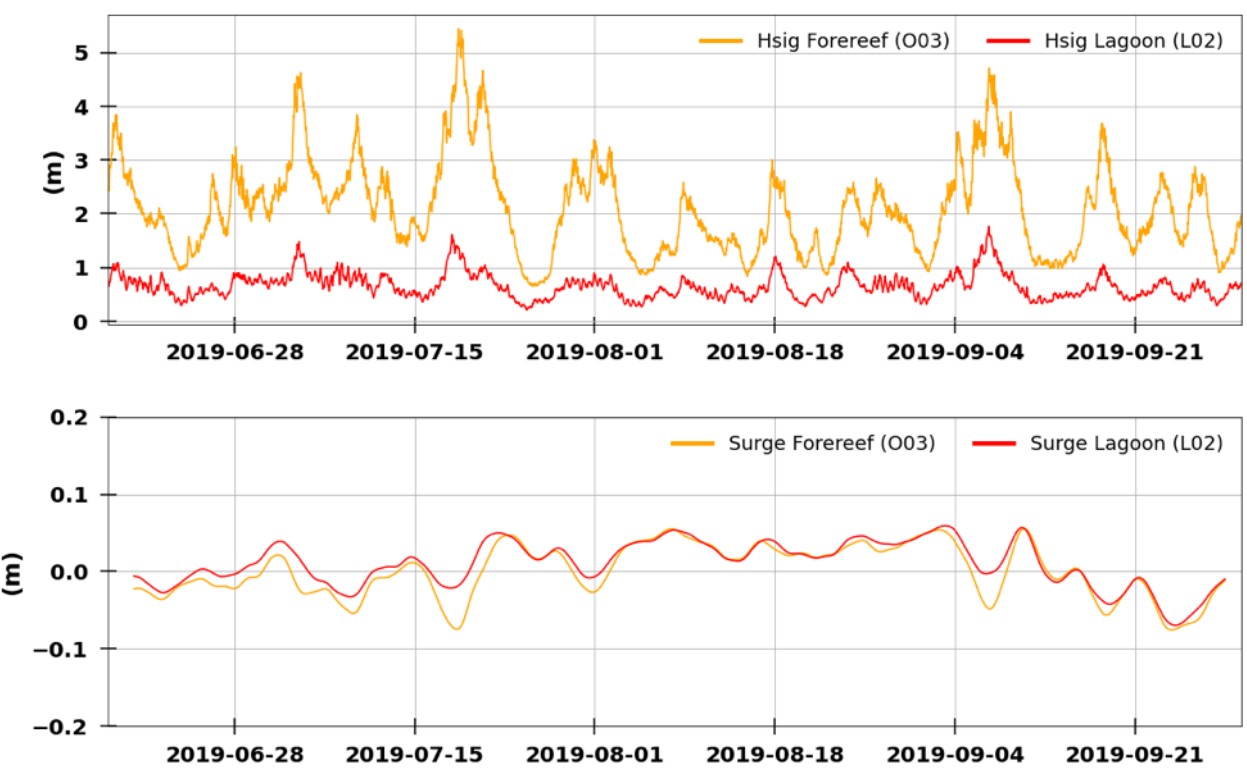

**Figure 8: Time series of Significant Wave Height measured in O03 and in lagoonal station L02 during Leg1 (upper panel). Surge (m) determined using Demerliac filter in O03 and L02 stations (lower panel).**

## 5.3 Temperature

Temperature in Gambier lagoon followed as expected a seasonal evolution, with lower temperatures in July (around 22°C) and a maximum reaching 28°C in February (Figure 9). During the austral winter season, lagoonal temperatures are lower than ocean by approximately 1°C. From late August to October, temperatures are similar and start to deviate from each other in November. Ocean is then cooler than the lagoon in summer period. During Leg 1, in austral winter, temperature appeared fairly stable, and rather cool, with values restricted between 22 and 24°C. Between June and August, three synchronous (all stations) temperature drops are observed on the entire water column (bottom and

surface) with temperature decreasing by 1 to 2°C depending on the location. Return to previous values
was more or less rapid depending also on the stations, with surface waters warming up faster than deep
areas. This phenomenon can be primarily related to intense wind events (Figure 3) or wave events
(Figure 6-7).



**Figure 9:** Time series of daily lagoon's temperature measured with SBE56 data loggers at surface and bottom depths (respectively
approx. 3m and 20m) along Leg1 and Leg2 deployments. Oceanic temperature measured at O02 station is shown on top panel in
red colour. Positive or negative temperature differences between surface and deeper areas are respectively highlighted in blue and
black.



The overall temperature dynamic during Leg2 shows an increasing trend starting November 2019 until February, and a delta >5°C is observed across the period. Similar to Leg1, there are however several
temperatures drop events synchronous for all stations, in particular in December 2019. The drop is recorded by all lagoon stations, from 2°C for L01, L02 and L03 to 2.5°C in L04 in 6 days, and can be assigned to periods of high southeast winds.

## 5.4 Currents and lagoonal circulation

Depth-averaged water circulation observed in Gambier lagoon shows that currents are dominated first by tide, and modulated by wind orientations and intensity. The ADCP_3 for instance recorded a depth-averaged symmetric direction-oriented NW-SE related to the natural alignment of the channel formed between Mangareva and Taravai Islands and the regular effects of tide (Figure 10). The same directional symmetry occurs in ADCP_1 (Figure 10).

Rikitea lagoon is influenced by water movements on its three sides. While in and outflows at ADCP_4 are balanced and seem primarily dominated by tide, inflows are more dominant in Aqua_2, due to the effects of both tide and southeast wind (Figure 3). Conversely, outflows are more persistent in Aqua_1 station, a feature that can interpreted as the outflow compensating the inflows elsewhere. The Rikitea sub-lagoon circulation has been specifically studied in Bruyère et al. (2023a).






**Figure 10: Map of depth-averaged current speeds (color legend in m/s) and direction recorded by current profilers (ADCP and Aquadopp) during the Leg2 deployment. Number of direction occurrences (%) are scaled according to the circles situated on the top right. Oceanographic convention is applied on current directions.**


Depth-averaged current speeds and directions measured from current profilers are presented in Figure 11. Measurements were restricted between 1st December 2019 to 12th December 2019, to highlight the influence of wind intensity and direction on currents. Indeed, a strong trade wind event (SE direction

and approx. 12 m/s) is observed between 7th December and 14th December (Figure 3). This event has been recorded by current profilers, Aqua2 is influenced by wind intensity and direction. In contrast, current directions observed in stations ADCP_1, ADCP_3 and ADCP_4 follow tide cycles but a small increase in speed is remarkable.




**Figure 11: Time series of depth-averaged current speeds (black line) and directions (polar legend) from ADCPs and Aquadopps instruments moored during Leg 2. Directions follow the oceanographic convention (direction where the current is going).**





## 6 Data availability

Data set is made publicly available through SEANOE open data publisher (https://www.seanoe.org/, Seanoe, 2023) in a dedicated repository. The registered database link to the following DOI https://doi.org/10.17882/94148, Andréfouët et al. (2023b).

## 7 Conclusion

The data presented in this paper provides a continuous times series of oceanographic data (currents, 365 temperature, water levels and waves parameters) recorded from June 2019 to February 2020 in Gambier Islands lagoon. This data set represents a first physical oceanography observatory for this site. Data were suitable for model validation as shown in Bruyère et al. (2023a). Those measurements offer the possibility to study the hydrodynamic features in relation to pearl farming activities, including spat collection (Bruyère et al., 2023a). For future work, *in situ* measurements could be extended to the 370 southern part of the Gambier lagoon as well as a specific investigation of the lagoon circulation during the winter season when swell events with high (>3m) wave height occur. Furthermore, the data set presented here can also be extremely useful for other investigations, beyond pearl farming. Namely, physical oceanography data can be helpful to understand the variability in occurrences of ciguaterra fish poisoning that has been a severe problem in Gambier lagoon (Chinain et al. 2016), biodiversity 375 resilience and larval recruitment for coral and invertebrate species other than oysters, and effects of land-born pollutants.

### Author contributions

**Oriane Bruyère** : Conceptualization, Writing - Original Draft, Visualization, Investigation Data Curation. **Romain Le Gendre** : Investigation Data Curation, Writing - Review & Editing. **Vetea Liao** : 380 Funding acquisition, Investigation, Writing - Review & Editing**. Serge Andréfouët :** Conceptualization, Visualization, Investigation Data Curation, Writing - Original Draft, Funding acquisition.

### Competing interests

The authors declare that they have no conflict of interest.

### Acknowledgements

385 The authors acknowledge the Direction des Ressources Marines (DRM) of French Polynesia for their financial support and for providing oceanographic instruments. The additional scientific staff that helped during the field operation described here include David Varillon, Bertrand Bourgeois, John Butscher, Manui Tanetoa and Teranui Ebb. We also thank the boat drivers and local population for their support and welcome.



## Financial support

This study was funded by a grant ANR16CE320004 MANA (Management of Atolls project). For Takapoto Atoll surveys were also funded by the Direction des Ressources Marines (DRM) through grant 7518/VP/DRM to IRD. Instruments were provided by the Direction des Ressources Marines, OTI project, Contrat de Projet France French Polynesia, Program 123, Action 2, 2015–2020

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
