# Peer review of "Lagoon hydrodynamics of pearl farming islands: the case of Gambier (French Polynesia)"

_Earth System Science Data, 2023_

## Author Comment (AC1)

**Reply to RC1 (Stuart Pearson) :**

We have taken care of and applied all minor comments and requests related to formatting or English language issues. We do not develop an answer for those requests. Below we provide answers in case of disagreement or when some details were asked.
Replies are show in Red font below.

**General Comments**

This manuscript presents a hydrodynamic dataset for the Gambier Islands (French Polynesia) in the Pacific Ocean, detailing waves, currents, temperature, and water levels at multiple locations around an atoll for 9 months in 2019-2020. The entire atoll seems well-instrumented, with instruments suitable to capture a wide range of hydrodynamic phenomena.

This article furthers ESSD's goals of clearly presenting and documenting a high quality earth science dataset. The authors' dataset is unique, capturing the hydrodynamics of a remote coral atoll in with relatively wide spatial coverage and fine temporal resolution. Such a dataset is not trivial to produce and will provide a valuable addition to the scientific literature. The dataset is also useful in itself for understanding atoll hydrodynamics, but has already apparently been used as a basis for model validation (Bruyere et al 2023b). I have made many comments below but I believe they are mostly minor in scope and in the spirit of helping a strong dataset and manuscript shine brighter.

As coral atolls are at high risk to the effects of climate change, such a dataset provides a welcome snapshot of a vulnerable location that may be valuable for researchers in other fields (e.g., ecology, coastal engineering) or studying other atolls. I can think of several colleagues who may be interested in using this dataset for their research on coral reef hydrodynamics (including myself), although it seems strange to filter out low-frequency wave data from this dataset if it was measured (see detailed comments below). As far as I can tell, the dataset here is complete and makes a coherent collection, and is sufficiently presented by this manuscript.

**Data quality**

The data is very clearly available in well-formatted netcdf files from the website. I downloaded several of the files to check the metadata and accessibility of the data, and it was very easy to find what I was looking for. I will actually share this with my students as an example of good practice for sharing research data.

Thank you.

However, I think that more attention to error/uncertainty estimates and processing procedures should be given in the manuscript (see detailed comments below). The current quality control section (Section 4.5) is only two sentences long and would benefit from more detail. There was not much discussion of errors or data cleaning/processing, beyond some mentions of unlinked Python datasets.

See below.

**Presentation quality**

The article is not too long and is clearly written, covering most of the major questions that I had when I started reading. The figures are all generally clear and well-formatted. Overall the presentation quality is good.

As per ESSD guidelines, "The authors should point to suitable software or services for simple visualization and analysis, keeping in mind that neither the reviewer nor the casual "reader" will install or pay for it." Perhaps then Section 6 (Data Availability) should here indicate again that the data is stored as NetCDF files and that they can be readily analyzed and visualized using a wide range of freely available code/software packages, or something along those lines.

Ok, added in the 'Data availability' section.

- L149-151: General question: how were your instruments mounted? E.g., were they placed on frames, laid directly on the seabed, bolted to a reef, weighted down? If placed on a metal frame, were your compasses calibrated and did you account for interference from the metal frame (e.g. for ADCP current direction)?

This is sensor dependent, but all instruments were tied to the bottom (not weighted) with aluminum or steel poles. Then, for all ADCPs we used the non-magnetic frames provided by the different constructors, while for the temperature and pressure sensors they were directly tied to a single pole that was hammered into the soft sediments (in lagoons) or into the hard reef matrix (on forereefs and passes). For ADCP, the holes in the frame prepared by the constructor to insert such poles were used, hence at some distance to the sensor. Standard deployment calibration procedures and parametrization (ENU coordinate system) were systematically used as per the constructor guidelines. The photos Figure 8 of the companion paper by Bruyère et al. 2023b (or essd-2023-198) shows these moorings.

We added in the text the mounts were non magnetic, and the possibility to check for essd-2023-198 to visualize the type of mounting.

- L167: Can you provide more detail on the processing ("processed with Python routines")? Is the code available online? If so, that would be good to mention here and link.

For data processing, we utilized standard Python libraries. In terms of data accessibility, most of the software employed for data access supports CSV conversion, simplifying the processing. However, for the RBR instrument, a specific package called "pyrsktools" was necessary to read the dataset, and this library is provided by the manufacturer. For processing the RBR instrument data, we employed the script outlined in Aucan et al. 2017. Lastly, for data conversion, we utilized the NetCDF4 package.

This has been specified in the revised text (now Line 172)

L178/184: How were your pressure/wave data processed? Can you provide more details?

We provided more details by inserting the text used in essd-2023-198 (Bruyère et al. 2023b) which refers in particular to Aucan et al. 2017, for even greater details on the processing.

- L182: Why did you use a constant atmospheric pressure to offset your wave gauges instead of measurements from the nearby weather station? What are the limitations of this assumption?

This is good point. However, this corresponds to a procedure more easily implemented anywhere (where there are no meteo stations), it also facilitates data reuse, with limited errors if the weather is 'standard', which was the case, In case of depression, a different procedure will have to be applied especially if there is reliable local weather station.

- L184: You only examine waves with periods 3-25 s, but infragravity (25-250 s) and very low-frequency (>250 s) waves are often very important on coral reef flats. Why did you make this choice? Is the lower-frequency data still available? This data would be extremely valuable for e.g. predicting flooding on atoll islands and validating existing theories about low frequency wave hydrodynamics on coral reefs. If you still have the raw data at these frequencies, I would strongly encourage you to add it to your dataset as I can think of quite a few people (myself included) who would find it useful.

This is also a good point, but to be honest, we did not consider infra gravity signals here on our investigations, or rather we did not aim to do so from the start, as we are not familiar with the analyses of the component. This is a growing subject but we have yet to drift that way. And Data Papers such as this are not meant to analyze all the data in every possible way.

We agree however it is something to keep in mind, for some applications. We added this perspective at the end of the paper, when mentioning other studies that could re-use the data set (and in this case with a different filtering procedure).

- Figure 5: Is there any temporal lag in the two signals? They look very close, so it's a bit hard to tell, but I think this is useful information.

We added information about the absence of temporal lag on line now 263 in relation to the Gambier Lagoon. We confirm that the tidal signals from both the ocean and the lagoon are synchronous. The Gambier Lagoon is highly connected to the ocean, which explains the similarity in tidal patterns between the ocean and the lagoon.

- L182: How long were your bursts? (e.g. 20 mins every hour)?

The duration of our burst varies depending on the set parameters. Some instruments were configured to record 180 pings per burst, which equates to 90 seconds. Then our burst occurred every 20 minutes during 90 seconds.

- L209: See my earlier comments re: compass calibration if you used a metal frame.

We specified the non-magnetic aluminum frames provided by the constructor were used. See Line now 216.

- L221: Again, is this code available?

See previous answer on similar request (and answers line now 172 and 192).

- L226: Please provide more detail on the data cleaning process.

We have already specified each step of our control procedure. However, we have added in line now 236 that incorrect sub-surface data of ADCP were removed from processing files.

- Section 7 (Conclusions): Something that I missed was a sentence or two describing how similar/different this location is to other sites (and therefore how can it be applied/useful for scientists/engineers at different locations besides the Gambier Islands)? Perhaps also circle back to the other French Polynesian sites mentioned in L43-53?

A comment has been added following the request on the new conclusion.

- Relatedly, a relatively limited pool of references was used, so I think the manuscript would benefit from a few additional references to other similar atoll measurements from other corners of the literature, such as:
  o Rogers, J. S., Monismith, S. G., Koweek, D. A., Torres, W. I., & Dunbar, R. B. (2016). Thermodynamics and hydrodynamics in an atoll reef system and their influence on coral cover. Limnology and oceanography, 61(6), 2191-2206.
  o Grimaldi, C. M., Lowe, R. J., Benthuysen, J. A., Cuttler, M. V. W., Green, R. H., & Gilmour, J. P. (2023). Hydrodynamic and atmospheric drivers create distinct thermal environments within a coral reef atoll. Coral Reefs, 1-14.

Actually these references are for atolls, while this site (Gambier) is not an atoll. Atolls instrumentations were described in the companion paper Bruyère et al. 2023b which was dedicated to atolls, now in press (or should be shortly).

**Technical Corrections**

- L27: Cite Bruyère et al (2023a) here just to make it clear that the biophysical model is in that paper and not in the current manuscript.

Done

- L53: Maybe add something here about human influences (or lack thereof) on the atoll? E.g. have the channels between islands been dredged or land reclaimed for runways etc? This sort of information could also be appropriate elsewhere in Ch. 2.

No, no such thing in Gambier.

- Figure 2: Indicate location of weather station (and ERA reanalysis point) on map with a dot and label?

Figure 2 has been modified.

- L100: Indicate what the vertical datum is? (e.g. measurements relative to mean sea level or some other local datum?)

For bathymetry or altitude, the reference should be the French Navy Hydrographic zero datum, although the 91m for the weather station cannot be ascertained, but this is the description provided by Meteo France. We added the reference next to the 91m.

- L100: I know it's in Figure 3 implicitly (and later in L124-125), but since this is the first time it comes up, I think you should explicitly state here in the text which dates your data spans during Leg 1 and 2.

Done

- L112: Clarify "…probably related to wave events." Is it worth mentioning the difference here between local wind waves and remotely-generated swell?

Clarified.

- L118 (and Figure 4): What is the (cross-shore) position on the reef? E.g., were you measuring on the shallow reef flats or deeper on the fore reef or in the lagoon etc?

All Oceanic ('O') sensors were moored on the forereefs as already specified in the text, while the other sensors are either on reef flats ('P' and 'Aqua') or on the lagoon ('L' and 'ADCP'), also as specified in the text. See the descriptions L130-140.

- Figure 4: Could you add a sentence or two to the text of the manuscript about how Andrefouet & Bionaz (2021) did the seabed classification? E.g. was it just based on depth or also on ecological parameters?

Please refer to the paper itself for the mapping procedure as it cannot be simply summarized in 2 sentences. It is also used here as a background just to illustrate the position of sensors and we believe it does not justify methodological developments.

- L224: RIP P01. I am impressed that you only lost one instrument during such an extensive campaign, well done!

Thank you

- L227: What is the range beyond which something was considered "out of range"?

In the case of this study, we did not observe any 'out-of-range', meaning any erratic data or clearly abnormal values in impossible range for the region, likely because our instruments were new and calibrated. Regarding temperature, we mainly filtered data above 40°C but this is arbitrary and based on local knowledge, and can be site dependent.

- L238: Capitalize "python" à "Python"; can you cite the T-Tide package?

We have added a citation of 't-tide' package in line now 260.

- Figure 6: Could you indicate the side of the atoll next to each station in the legend? E.g., "O03 (SE); O04 (N)", etc. I think this would help the reader with interpretation, especially with connecting the long-period swell vs local sea in panel (C) with your text in Section 5.2.

It has been added to the Figure 6.

- Highlight that this data could also be useful for flood hazard estimation or validation of early warning systems (see Winter et al…)

Mentioned now in conclusion, along with possible perspectives

- L324: the delta T described here is with respect to time for a single sensor across the full period (i.e. max dT/dt), rather than the max temperature difference between the surface and bottom during the period (i.e. max dT/dz), correct? Some clarification here would be helpful.

This has been clarified lines now 345-346

- L347: Given that you describe thermally stratified conditions in L314 and Figure 9, how appropriate is it to show depth-averaged flow here? Is there any shear in the velocity? Perhaps it is good to mention in ~L211-216 whether only the depth-averaged flow is included in the ADCP dataset or if you have provided all the bins.

This is a clever point, but Figure 9 is a representation which is not aimed to be coupled with stratification interpretation, but to illustrate general current conditions. We agree a different representation will be needed should an investigator would like to go this way with appropriate data representation. The full ADCP data set is provided, hence all the bins. This has been clarified in the sensor section.

END reply to RC1

---

## Author Comment (AC2)

**Reply to RC2 (Robert Schlegel) :**

We have taken care of and applied all minor comments, technical comments and requests related to formatting or English language issues. We do not develop an answer for those requests.
Below we provide answers in case of disagreement or when some details were asked.
Replies are show in Red font below.

**General comments**

**Summary**

The authors document the design, deployment, and curation of an array of oceanographic motioning instruments and the various data streams that they generated. This was done specifically within and around the lagoons of Gambier Islands, French Polynesia for the purpose of better quantifying the processes that are likely having an (adverse) impact on oyster spat. Though as the authors point out, such a wide array of high temporal resolution data can be used for many different investigations. The authors quickly move through the background to the project, the deployment/retrieval, QC, and a description of each type of data. Providing insights into the importance of the findings as they go. The conclusion is rather short, but that is fine. As the authors also point out that more in-depth studies utilising these data have already been (or are in the process of being) published.

The effort needed to manage a project of this scope is commendable and the dataset (as a series of NetCDF files) is well packaged and openly accessible online. The data are easy to download, extract, and work with. Though I think it would have been better to package the data into one single NetCDF file. Or at least just one NetCDF file per instrument type. For example, it puts unnecessary encumbrance on the user to have to separately download, load, and combine files for L01_3m and L01_21m when the only difference between them is depth, which is already one of the attributes in the NetCDF file. By creating a spray of files like this it reduces the R of the data (i.e. Reusability; FAIR). That being said, the data are already published and I don't think it's necessary that they be recombined, even though it would be of some benefit to the community, and to the posterity of the data themselves.

We understand this reasonable suggestion, although by experience, some users have requested data from only one sensor, hence the choice to maximize the granularity and avoid multi-sensors files.

The authors are perhaps a bit heavy on their use of figures. And the contrast between colour palettes and aesthetic styles between figures is sometimes a bit jarring. I did however enjoy most of the figures and found some of the visualisations to be quite interesting. I assume that some of the figures were made with different software applications, which will prevent the authors from maintaining a consistent aesthetic throughout. So I would recommend trying to have at least a consistent font face, and/or develop a border for the figures in post-processing that looks the same in order to ensure a more contiguous visual experience for the reader. And please replace rainbow colour palettes whenever possible. The 'viridis' colour palettes being one easy choice.

Not sure why the rainbow palette is not adequate here. The first reviewer is not mentioning this as a problem. Since it was not a mandatory requirement to change it even from the reviewer (see below), we have left the initial color palette. We also confirm to you that all our figures

were created using Python software, except for Figure 10, which was generated using GIS software.

Other than that, my only recurring criticism is the quality of the writing. The text would benefit from being passed through a language correction software like deepL in order to bring it closer to a native English level. I was however able to progress through the manuscript and understand the authors meaning for every sentence on the first read (which I find is often not the case for physical oceanography papers). So I am not suggesting that any structural re-writing is necessary.

Thanks. English is not our first native language, so it will never be at the level of native English speaker. As far as the scientific message is perfectly clear, this is ok for us.

Overall I think the paper effectively communicates what data were collected, why they were collected, what they look like, and where to find them. Whereas I do have some minor points below, these are just comments on the grammar/syntax before I stopped editing the language too closely. I think the authors should re-consider how they visualise their data, but I don't think this is absolutely necessary for this manuscript to be published. Therefore there are only minor revisions to make.

**Specific comments**

**Data availability**
- Link to data works well. Data can be downloaded and are in a standard NetCDF format. Though I wasn't sure what the 'CORRELATION' values were meant to show in the ADCP files.
Although we can not enter such detail sin the paper itself, CORRELATION is the signal to noise ratio used as a measure of data quality, among 4 critical variables that are included in the NetCDF output (Correlation, Velocity, Percent Good and Echo Intensity), based on the parameterization used. More information can be provided in various documentations, a complete useful one from a user's perspective can be found here https://pubs.usgs.gov/of/2000/of00-458/pdf/ofr200-458toolboxmanualv4_508old.pdf

It would be preferable if the code used for the analyses / data QC were also made publicly available (e.g. GitHub etc.).

See reponse to Reviewer 1 on this code availability question. Line now 172 of the revised text.

**Table 1**
It's very impressive to see how much work this is when summarized into a table.
I recommend putting the rows of moored instruments only for Leg1 at the top of the table.
Ok, done.

**Figure 1**
Very nice. I like the layout of the progressively zoomed in map panels. Nice attention to detail how there is more and more colour as the panels zoom in. I recommend removing it and just having the one land mass shown.
We are not sure what is recommended, and where, here in term of 'one land mass shown'? Is for Figure 1 or 2?

Note that we changed the reference to the Sentinel 2 image, in agreement with comments made on paper essd-2023-198 (or Bruyère et al. 2023b)

**Figure 2**
I don't think the inset is necessary. It's not much more zoomed in than the main image.
Ok, modified.

**Figure 3**
Rather use a non-rainbow colour palette to show wind direction. Such as one of the viridis colour palettes. "Wind directions follow meteorological convention" Meaning that the direction shown here indicates that is where the wind is coming from, or towards where it is going?

We have not modified the color palette as it was not mandatory, and also to keep the same palette as in other publications (Andreofuet et al. 2023a, Bruyère et al. 2023a). However, we have tried (below), but we don't like it as much as the rainbow palette!
For wind, the direction is where the wind is coming from (hence 0° is a wind coming from the North).

[Figure]

**Figure 4**
"…" -> "."

**Figure 8**
Add names to y-axes, and panel labels (i.e. (A) (B) )
Ok done.

END reply to RC2